


**Investigating the importance of sub-grid particle formation in point source plumes over eastern China using IAP-AACM with a sub-grid parameterization**

Ying Wei[1,2], Xueshun Chen[2,3,*], Huansheng Chen[2], Yele Sun[2,3,5], Wenyi Yang[2,3], Huiyun Du[2], Qizhong Wu[4], Dan Chen[1], Xiujuan Zhao[1], Jie Li[2,3], Zifa Wang[2,3,5]

[1] Institute of Urban Meteorology, China Meteorology Administration, Beijing, 100089, China

[2] The State Key Laboratory of Atmospheric Boundary Layer Physics and Atmospheric Chemistry, Institute of Atmospheric Physics, Chinese Academy of Sciences, Beijing 100029, China

[3] Center for Excellence in Regional Atmospheric Environment, Institute of Urban Environment, Chinese Academy of Sciences, Xiamen 361021, China

[4] College of Global Change and Earth System Science, Beijing Normal University, Beijing 100875

[5] University of Chinese Academy of Sciences, Beijing 100049, China

Correspondence: Xueshun Chen (chenxsh@mail.iap.ac.cn)

**Abstract:**

The influence of sub-grid particle formation (SGPF) in point source plumes on aerosol particles over eastern China was firstly illustrated by implementing a SGPF scheme into a global-regional nested chemical transport model with aerosol microphysics module. The key parameter in the scheme was optimized based on the observations in eastern China. With the parameterization of SGPF, the spatial heterogeneity and diurnal variation of particle formation processes in sub-grid scale were well resolved. The SGPF scheme can significantly improve the model performance in simulating aerosol components and new particle formation processes at typical sites influenced by point sources. The comparison with observations at Beijing, Wuhan, and Nanjing showed that the normal mean bias (NMB) of sulfate and ammonium could be reduced by 23%-27% and 12%-14%, respectively. When wind fields were well reproduced, the correlation of sulfate between simulation and



observation can be increased by 0.13 in Nanjing. Considering the diurnal cycle of new particle formation, the SGPF scheme can greatly reduce the overestimation of particle number concentration in nucleation and Aitken mode caused by fixed-fraction parameterization of SGPF. In the regional scale, downwind areas of point source got

an increase of sulfate concentration by 25%-50%. The results of this study indicate the significant effects of SGPF on aerosol particles over areas with the point source and necessity of reasonable representation of SGPF processes in chemical transport models.

**Key words:** IAP-AACM, sub-grid particle formaiton, secondary inorganic aerosol,

particle number, China

## 1. Introduction

Air pollution caused by high concentrations of aerosol particles has become a primary environmental problem in metropolis and caused widespread public concern in China (Zhang and Cao, 2015; Sun and Chen, 2017). Atmospheric aerosol particles

affect atmospheric visibility and public health while also has significant climatic and ecological effects (Zhang et al., 2010; Boucher et al., 2013; Powell et al., 2015), which are closely related to their size distribution and chemical composition (Spracklen et al., 2005; Dusek et al., 2006). As one of the main inorganic aerosol component, sulfate directly changes the energy budget of the earth-atmosphere system

by scattering solar radiation, and affects the climate acting as cloud condensation nuclei (CCN) (IPCC, 2013). Sulfuric acid ($H_2SO_4$) is the core material in the nucleation and growth of particles. The hygroscopicity of sulfate triggers heterogeneous reactions of gas precursors by changing the aerosol water content (Zhuang et al., 2014). Sulfate can also enhance the extinction and contribute to haze

pollution by mixing with other components (Zhu et al., 2010; Zanatta et al., 2018), affecting the formation of secondary inorganic aerosol (Adams et al., 1999) and secondary organic aerosol (SOA) (He et al., 2018). Coal consumption in China contributed about 80% of sulfur dioxide ($SO_2$) emissions over the last decade, with 70%–90% being from power plants and industrial emissions (Wang et al., 2014; Ma et



al., 2017; Zheng et al., 2018). Thus $SO_2$ from energy and industrial sectors constitutes the major source of sulfate in China.

For point sources, the oxidation from $SO_2$ to sulfate is a typical sub-grid phenomenon with a more rapid conversion rate in the plume. In the sulfur-rich plume, a swiftly gas-to-particle process occurs through the reaction $2OH^- + SO_2 \rightarrow H_2SO_4$

(Kulmala and Kerminen,2008), due to the higher concentration of nitrogen oxides ($NO_x$) and volatile organic compounds (VOCs) in the plume than in the ambient atmosphere, and also due to the inhomogeneous meteorological condition (i.e., temperature, relative humidity (RH)) and aerosol concentration within and external to the plume (Yu, 2010; Lonsdale et al., 2012; Stevens et al., 2012). $H_2SO_4$ can either

condense onto pre-existing particles or nucleate to form new particles, which are treated as primary sulfate, or so-called sub-grid sulfate (SG-$ASO_4$). A significant increase in both mass and number concentrations of particles has been observed downwind of coal-fired power plants (Richards et al., 1981; Gillani et al., 1998; Brock et al., 2002). Yu (2010) showed that particle formation in the plume depends on

hydroxyl radicals (OH) concentration, which varies diurnally. The fraction converted to sulfate thus has a clear diurnal variation, leading to spatial-temporal heterogeneity in local particle mass concentration and particle number size distribution (PNSD).

The modeling distributions of both particle number and component mass are sensitive to the fraction of $SO_2$ oxidized to sulfate in the plume. Currently, for

SG-$ASO_4$, an average proportion of 0%–5% of all $SO_2$ emissions is often considered to reflect the $H_2SO_4$ emitted to every grid in chemical transport models (CTMs) (Textor et al., 2006). Furthermore, the fraciton of $H_2SO_4$ that forms new particles through nucleation is taken as 0%–15% in aerosol microphysical models (Luo and Yu, 2011; Chen et al., 2018). The assumption of an averaged fraction of oxidation and

nucleation in the grid neglects the diurnal cycle of sub-grid particle formation (SGPF). and furthermore cannot capture the spatial-temporal variation of particle formaition in the plume, nor does it account for the effect of $H_2SO_4$ condensing onto pre-existing particles, which may have a significant impact on PNSD. A simulation in North China employed 26.5% of primary fine particulate matter as SG-$ASO_4$ indicated a monthly



averaged contribution to sulfate of ~10%–20% (Zhang et al., 2012). The assumed

proportion of SG-ASO$_4$ caused an uncertainty in global CCN concentration of up to

40%, and over 100% in polluted regions (Spracklen et al., 2005). As SGPF may occur

on a spatial scale of tens seconds of kilometers, this leads to large uncertainties in

predicting the spatio-temporal variation of particle number and mass caoncentration

during dilution in the plume when it is calculated as a grid-averaged concentration, as

in most CTMs (Spracklen et al., 2005, 2008).

To solve the uncertainties caused by grid-averaged fraction of SO$_2$-to-H$_2$SO$_4$, a

plume-in-grid (PinG) model coupling the smoke plume model into the Eulerian Model

has been developed in air quality models such as the Community Multiscale Air

Quality model, the Comprehensive Air quality Model with extensions and the Weather

Research and Forecasting/Chemistry (Gillani et al., 1999; Karamchandani et al., 2002,

2010). However, the implementation of PinG greatly increases the computational

burden due to the large amount of fine grids resolved. Due to this reason, PinG is not

suitable for complicated aerosol models involving microphysical processes. Stevens et

al. (2013) developed a computationally efficient sub-grid parameterized scheme, the

'Predicting Particle Production in Power-Plant Plumes' (P6) scheme, based on

physicochemical processes of particle formation. The scheme was incorporated in a

global CTM with aerosol microphysics module and the sub-grid effect on particle

number concentration was evaluated against observations over North America and

Europe. However, urban sites were excluded from the evaluation and the sub-grid

impact on aerosol components was not described. There are, in fact, few modeling

studies involving sub-grid particle characteristics in high polluted regions such as

China. The severe air pollution in China means that atmospheric chemical

characteristics there are quite different to those in other countries. Higher atmospheric

oxidability and particle growth rates have been reported in recently researches (Wang

et al., 2017; Tan et al., 2019; Yang et al., 2020; Liu et al., 2021). This means OH

concentration (key parameter of the oxidation process) parameterized by NO$_x$

concentration in the P6 scheme is not suitable for the atmospheric condition in China,

and the characteristics of SGPF in plumes should be different here.



In this study, we coupled the P6 scheme to a global-regional nested atmospheric chemistry model with an aerosol microphysics module to better describe the process of SGPF in plumes. Moreover, the localization of the sub-grid scheme (refers to SGPF scheme) was carried out based on the observed high level of radicals caused by the polluted background in central–eastern China. With the updated model, the

improvements in simulating aerosol composition and PNSD were evaluated by comparing with abundant observations in eastern China. The original model and its updating are described in Sections 2.1–2.3. Simulation experiments and observations are introduced in Section 2.4 and 2.5, respectively. Meteorological fields are verified in Section 3.1. The evaluation and improvements to the updated model against

observations are desctribed in Section 3.2–3.4. The influence of SGPF on regional scale is analyzed in Section 3.5.

## 2. Methods

### 2.1 Description of IAP-AACM

         The Aerosol and Atmospheric Chemistry Model of Institute of Atmospheric

Physics (IAP-AACM) is a multi-scale nested three-dimensional chemistry transport model coupled to the Earth System Model of the Chinese Academy of Sciences (CAS-ESM) (Wei et al., 2019; Zhang et al., 2020). The IAP-AACM was developed on the basis of the Nested Grid Air Quality Prediction Model System (NAQPMS) (Wang et al. 2006b) and the Global Nested Grid Air Quality Prediction Model System

(GNAQPMS) (Chen et al., 2015). NAQPMS/GNAQPMS are widely used in the simulation of dust (Li et al., 2012; Wei et al., 2019), ozone (Wang et al., 2006a; Li et al., 2007), deposition (Ge et al., 2014), air pollution control policy (Wu et al., 2011; Li et al., 2016; Wei et al., 2017) and global transportation of mercury (Chen et al., 2015). In the IAP-AACM, dimethylsulfide, sea salt and dust emissions are calculated online.

The dust scheme originates from the wind erosion model developed by Wang et al. (2000) and improved by Luo et al. (2006).

         The gas phase chemistry is calculated with the Carbon-Bond Mechanism Z model (CBM-Z; Zaveri and Peters, 1999). The photolysis rate is calculated mainly



considering altitude, latitude, longitude and the effects of clouds. The rates of
photolysis reactions depend on the spectral actinic flux and the spectral actinic flux
depends on the absorption and scattering of incident solar radiation by gaseous
molecules, clouds and aerosols. The photolytic rate constants typically increase with
height due to the reduction in the total integrated optical depth (OD) with lower
pressure, less aerosols and clouds (Seinfeld and Pandis, 2012; Williams et al., 2012).

The gas-phase chemical mechanism has important impacts on $NO_x$ and ultimately the
OH concentration calculated in the SGPF scheme. Different mechanism may have
different impacts on the parameterization result. Zhang et al. (2012) compared
simulations conducted in summer with three different gas-phase mechanisms (i.e.,
CBM-Z, CB05 and SAPRC-99) in WRF-Chem and found that simulations with all
three gas-phase mechanisms well reproduced the surface concentrations of $O_3$, CO,
$NO_2$, and $PM_{2.5}$. Prediction discrepancies caused by different mechanisms were of
mass concentrations of $O_3$ (up to 5 ppb), $PM_{2.5}$ (up to 0.5 $\mu g\ m^{-3}$), secondary
inorganic $PM_{2.5}$ species (up to 1.1 $\mu g\ m^{-3}$). Overall, the simulation discrepancy
between model with the CBM-Z mechanism and other widely used new mechanisms
should be reasonably acceptable.

The aqueous chemistry and wet deposition scavenging is simulated with the
Regional Acid Deposition Model chemical mechanism (Stockwell et al., 1990). The
heterogeneous chemistry uses the scheme described by Li et al. (2012). For aerosol
microphysical processes, the IAP-AACM in this study describes the size distribution
of aerosol particles using the Advanced Particle Microphysics (APM; Yu and Luo,
2009) module, as reported in previous studies (Chen et al., 2014; Chen et al., 2018).
The APM in IAP-AACM uses 40 sectional bins to represent secondary particles
formed from nucleation and subsequent growth with dry diameters of 0.0012–12 $\mu m$.
Black carbon (BC) and organic carbon (OC) particles are represented by 28 bins. Sea
salt and dust particles are represented by 20 bins and 4 bins, respectively. The APM
assumes these particles are the cores of particles and they are coated with secondary
species. Both the cores and coating species are tracked in the model. Semi-volatile
aerosol species, including nitrate, ammonium and SOA, are simulated by the bulk



method. Only their total mass concentrations are tracked and the concentrations

apportioned to particles in different sizes are assumed to be proportional to the mass

concentration of associated sulfate. Nitrate and ammonium are simulated by

ISORROPIA version 1.7 (Nenes et al., 1998, 1999). SOA concentration is calculated

with the scheme described by Strader et al. (1999). A comprehensive evaluation of the

simulation of IAP-AACM from global to regional was shown in Wei et al. (2019).

**2.2 Implementing sub-grid scheme into IAP-AACM**

In the original version of the IAP-AACM, $H_2SO_4$ was emitted directly into grids

in a fixed proportion with 5% being emitted into nucleation mode and 95%

condensing onto the existing accumulation mode particles in the APM module (Yu

and Luo, 2009). We updated the IAP-AACM+APM with the P6 sub-grid particle

parameterization scheme (Stevens and Pierce, 2013) to resolve the dynamic variation

of SGPF in the global nested model.

The P6 scheme includes the rapid conversion from $SO_2$ to sulfate within the

plume, considering both computational efficiency and physical basis. The training

data for constructing the P6 scheme are based on results of the large-eddy

simulation/cloud-resolving model named System for Atmospheric Modeling (SAM)

(Kairoutdinov and Randall, 2003) with a flexible resolution of tens of meters to

several kilometres. The model results used to construct the P6 scheme have been

tested against aircraft observations(Lonsdale et al., 2012; Stevens et al., 2012). For

more information on the model, refer to Stevens et al. (2012).

The sub-grid scheme resolves SGPF into two key processes, namely oxidation

and nucleation, involving parameters from the meteorological field, emission source

and environmental background as inputs. Accordingly, the oxidation of $SO_2$ emitted

from a point source is constructed with meteorological conditions (i.e., wind speed

($v_g$), boundary layer height (BLH), downward shortwave radiative flux (DSWRF)),

emissions of $SO_2$ and $NO_x$ ($NO_x$emis) from the source, mean background

concentrations of $SO_2$ and $NO_x$ ($bgNO_x$), and the distance from the source ($d$).

Nucleation of $H_2SO_4$ in the plume is constructed with factors mentioned above and





the mean background condensation sink. A scaling factor is used to allow the equations to fit the data when calculating the effective concentrations of $NO_x$ and $SO_2$ within the

plume, based on the reality that the emitted fluxes and the resulting of $NO_x$ and $SO_2$ concentrations in the plume are very different to the grid-mean values. A detailed calculation of the parameteriazation scheme is provided by Stevens et al. (2013).

Oxidation and nucleation process in the P6 scheme were integrated into the chemical reaction and aerosol microphysics modules, respectively. The key parameter predicted

during the oxidation process is the oxidation fraction of the emitted $SO_2$ ($f_{ox}$). The P6 scheme is combined with the gas-phase chemistry module to describe the dynamic variation of $H_2SO_4$ production within and outside the plume, based on the variation of meteorological conditions and environmental backgrounds. Emissions separated by sector is used in IAP-AACM to analyse the impact of different emission sources. We

used real-time online calculation of $f_{ox}$ to repartition sub-grid $H_2SO_4$ production and $SO_2$ from energy and industrial sector sources.

For the nucleation process, the $H_2SO_4$ produced in oxidation process is distributed to nucleate to new particles and condense onto pre-existing particles by the key parameter of new particle formation fraction ($f_{new}$). In the APM, the condensation of

$H_2SO_4$ onto pre-existing particles (sulfate, BC, OC, dust and sea salt) are proportional to the corresponding ratio of condensation sink of particles. Furthermore, the $H_2SO_4$ nucleating to NFPs is partitioned into 40 bins of secondary particles in the APM with a lognormal distribution as follows:

$$n_N(D_p) = \frac{dN}{dD_p} = \frac{N}{(2\pi)^{1/2} D_p \ln \sigma_g} \exp(-\frac{(\ln D_p - \ln \overline{D}_m)^2}{2\ln^2 \sigma_g}) \qquad (1)$$

where $D_p$ and $N$ represent the particle size and total number of particles, respectively; $\sigma_g$ is the geometric standard deviation of the aerosol size distribution of 1.4 in this study; and $D_m$ is the number-median diameter (μm) calculated by

$$D_m = (\frac{M_m}{\rho} \frac{6}{\pi})^{\frac{1}{3}} \exp(-1.5\ln^2 \sigma_g) \qquad (2)$$

where $M_m$ is the the mean mass of per newly formed particles (kg) predicted by the P6

scheme; $\rho$ is the density of dry aerosol (1.7 g cm$^{-3}$ in our model). After this





repartitioning, the tracers of secondary particles are updated through moving mass concentrations of sulfate across bins in the APM module.

**2.3 Optimization of the key parameter in the sub-grid scheme**

Since OH is very important for the diurnal cycle of SG-ASO4 conversion in the plume, the determination of OH concentration is crucial to the sub-grid scheme. In parameterization of the oxidation process, the key step is calculating $f_{ox}$ in the plume, which depends on the rate constant, $k$, time elapsed, $t$, and the effective OH concentration in the plume, $OH_{eff}$, [molecules cm$^{-3}$]. $OH_{eff}$ is calculated by:

$$OH_{eff} = 0.82 \cdot 10^{P1 \cdot \log(P2)/6.8} \qquad (3)$$

where P1 and P2 are the function of the effective NO$_x$ concentration in the plume (NO$_{x,eff}$, [ppb]) and the DSWRF, respectively, constructed from excessive training data. P1 and P2 are calculated as:

$$P1 = -0.014x^6 + 0.0027x^5 + 0.1713x^4 - 0.0466x^3 - 0.7893x^2 - 0.1739x + 6.9414 \quad (4)$$

$$P2 = (-1345y^3 + 4002y^2 - 471.8y + 42.72) \times 10^4 \qquad (5)$$

$$y = \frac{DSWRF}{S_0 \cdot T} \qquad (6)$$

where S0 is the solar constant at the top of the atmosphere, 1370 W m$^{-2}$, and T is an assumed transmittance of the clear atmosphere, 0.76. The relationship between DSWRF and OH concentration in SGPF scheme was displayed in Fig. S1. The simulated OH concentration is under $1 \times 10^6$ cm$^{-3}$ when DSWRF varies between 0-200 W m$^{-2}$. Thus, the fraction of total SO$_2$ emitted converted into particles in the plume much lower in cloudy days.

In the original P6 scheme, x was calculated from NO$_{x,eff}$ by:

$$x = \log([NO_x, eff]) - 0.195 \qquad (5)$$

where NO$_{x,eff}$ is related to NO$_x$emis, bgNO$_x$, BLH, $v_g$ and $d$ as mentioned in Section 2.2. for the low-VOC case for the isoprene mixing ratio of <0.15 ppb, or:

$$x = \log([NO_x, eff] \times 0.6) - 0.195 \qquad (6)$$

in the high-VOC case with the isoprene mixing ratio of 1.5 ppb. The effect of VOC



within the plume is not explicitly taken into account in the parameterization of $OH_{eff}$.

Stevens et al. (2012) discussed the potential effect of low and high VOC concentrations on OH production and related it to the concentration of $NO_x$ that corresponds to the peak concentration of OH. The large amount of oxidation of isoprene has material effect on peroxy radicals and tends to shift the peak of OH production to a higher level. It is reasonable to expect that the true peak OH concentrations should be shifted to a higher $NO_x$ than suggested by the P6 scheme.

The upper limit of the background concentration of $NO_x$ in the P6 scheme is ~8 ppb (Stevens and Pierce, 2013).

The $NO_x$ concentration in eastern China is commonly at a high level of >20 ppb, so there will be an underestimation of OH production in polluted regions if the original P6 scheme is applied directly. We therefore adjusted the OH concentration

parameter in the sheme to take into account the high oxidizability in the polluted atmosphere in eastern China. Although the $OH_{eff}$ calculated in the SGPF scheme is independent of the OH concentration calculated by CBM-Z and only used to calculate the $SO_2$ oxidation fraction to sulfate in the plume, the grid-averaged OH concentration calculated by CBM-Z has large imparct on $NO_x$ and VOCs and ultimately changes

$OH_{eff}$. The simulation of $NO_2$ and OH in the base model has been validated in Wei et al. (2019). Overvall, the model well reproduced the seasonal variation of $NO_2$ and captured the daily variations with R of 0.49-0.7 in most cities in eastern China. The global distribution of OH concentration in the IAP-AACM is similar with other CTMs. In addition, the simulations of the IAP-AACM with observations took at rural sites

located in the North China Plain near Beijing (Tan et al., 2017) and the Pearl River Delta near Guangzhou City (Lu et al., 2012) in summer were compared in Fig. S2. Although the observation in Guangzhou was took in 2006, the characteristics of atmospheric oxidation under the regional atmospheric background can still be roughly reflected. As shown in Fig. S2, the model reproduced the diurnal characteristics of OH

concentration well from south to north of China in summertime.

Field observations in North China indicate a considerable underestimation of OH levels in simulations for metropolitan areas. For warm seasons, Lu et al. (2013)



reported daytime peak OH concentrations of $(4–17)\times10^6$ cm$^{-3}$ at a suburban site in

Beijing in summer 2006. Tan et al. (2017) observed a daily OH maxima of $(5–15)\times$

$10^6$ cm$^{-3}$ at a rural site in the North China Plain in summer 2014. In cold seasons, the

radical concentrations are expected to be much lower than in summertime due to

limited photochemistry, but high concentrations are still found in recently-measured

records. Tan et al. (2018) observed the average daytime maximum OH concentrations

of $2.5\times10^6$ cm$^{-3}$ at another suburban site in Beijing in January 2016. Slater et al.

(2020) found daily maximum OH concentrations of $(1–8) \times 10^6$ cm$^{-3}$ with a daytime

peak of $2.7 \times 10^6$ cm$^{-3}$.

In view of these observations, we changed the coefficient of $[NO_{x,eff}]$ to 0.3 in the

scheme as shown in Eq. (7) to avoid an underestimation of SGPF, on the basis of

observations of OH and $NO_x$ in eastern China:

$$x = \log([NO_x, eff]\times0.3) - 0.195 \qquad (7)$$

The $NO_x$–OH concentration parameterization in both the original scheme and the

localized scheme at the site of IAP in Beijing is illustrated in Fig. 1. In the updated

parameterization, the maximum annual averaged OH concentration is still $\sim7\times10^6$ cm$^{-}$

$^3$, but the value of OH concentration corresponding to an $NO_x$ concentration of 15–30

ppb is revised from $(0.5–1.5)\times10^6$ cm$^{-3}$ to $2–4\times10^6$ cm$^{-3}$.

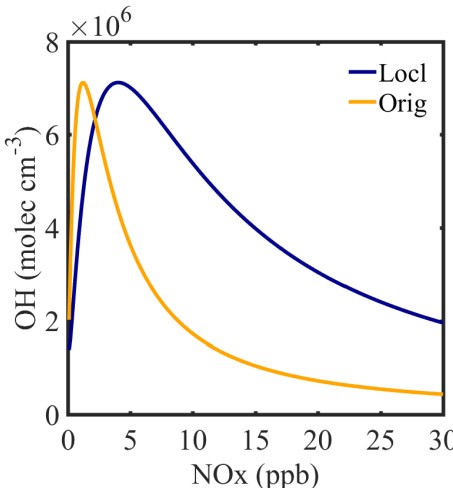

Fig. 1 the localized (blue line) and original (yellow line) parameterization of the





NO$_x$-OH concentration curve in the P6 scheme. The input parameters used to calculate the curve were annually averaged.

**2.4 Simulation design**

In this study, a nested domain over China with 0.33°×0.33° resolution was implemented to analyze the impact of SGPF on aerosol pollution in China, with the first domain covering the globe at 1°×1° resolution. Vertically, the model uses 20 layers, from the bottom layer centered at 50m to the top layer at ~20km, with 10 layers below 3 km. Meteorology was driven by a global version of Weather Research and Forecasting version 3.7 (WRF v3.7). The WRF was driven by the National Centers for Environmental Prediction Final Analysis (FNL) datasets with the calculation nudged to FNL data. The input frequencies are 3 hourly in the global domain and 1 hourly in the nested domain. The top boundary conditions for ozone, NO$_x$ and carbon monoixde were prescribed by the Model for Ozone and Related Chemical Tracers version4 (Emmons et al., 2010). For model performance evaluation and analysis of aerosol components, several cases corresponding to observation periods in 2014 (shown in Section 2.5) were simulated with one month for spin-up time. One simulation with the SGPF scheme (SG) and a control experiment with constant SGPF of $f_{ox}$=0% (F0) were conducted in the simulating period. The fraction of $f_{ox}$=0% represents the simulation without sub-grid particles. In addition, we conducted two simulations (with and without the SGPF scheme) for winter 2016 (described in Section 2.5) to evaluate model performance in PNSD at a typical urban site. The simulation without the SGPF scheme employed the $f_{ox}$=2.5% (F2.5) which refers to the AeroCom recommendation by Dentener et al. (2006). The simulating of F2.5 was also implemented in the comparison of the diurnal characteristics of SGPF in January and July 2014 in Section 3.2.

A global emission dataset of source categories (with 29 species and 14 sectors) was applied with anthropogenic emissions from Hemispheric Transport of Air Pollution version 2. Detailed information for the emissions is available from Wei et al. (2019). The SO$_2$ oxidated to SG-ASO$_4$ was from emergy and industry sectors (shown



in Fig. 2) , and was emitted into the first five and three layers of the model. The emissions in China were scaled to the level of 2014 based on emission trends reported by Zheng et al. (2018). The emissions of China for the simulation of 2016 were

updated to the Multi-resolution Emission Inventory for China-2016 published by Tsinghua University (Zhang et al., 2019).

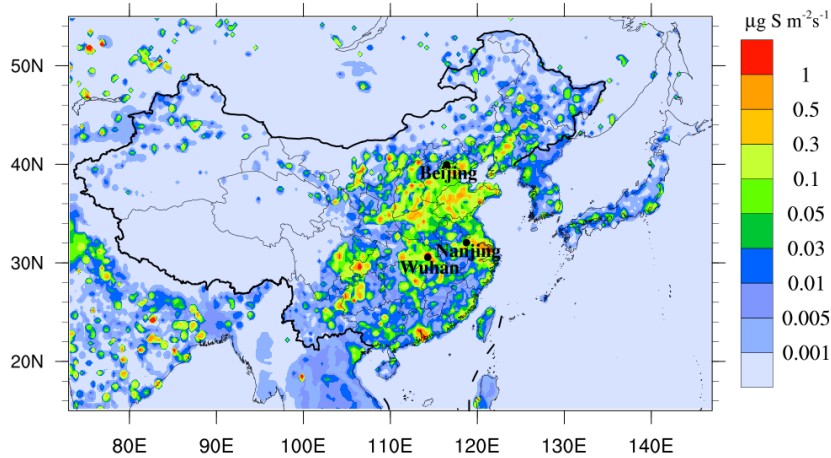

Fig. 2 The nested domain with annual mean $SO_2$ emission flux (unit : $\mu g\ S\ m^{-2}s^{-1}$) from energy and industry sectors in 2014. The black circles are locations of the

observation sites.

### 2.5 Observation data

The observations of aerosol components and PNSD were obtained from the Atmospheric Pollution and Human Health in a Chinese Megacity (APHH-Beijing) campaign conducted at an urban site (the meteorological tower of IAP) in central

Beijing during November–December 2016. Particle size ranges of 3–25, 25–100, and 100–1000 nm were applied for the nucleation mode, Aitken mode and accumulation mode, respectively. In addition, we collected the mass concentrations of sulfate-nitrate-ammonium (SNA) at urban sites in the central of Nanjing and Wuhan (see in Fig. 2) to evaluate model performance in simulating aerosol components. The

Nanjing and Wuhan sites were respectively located on the east and west banks of different reaches of the Yangtze River, with several power plants being located to their northeast. Locations and observation periods are given in Table 1. Observations to test



the meterological fields were collected from the National Climate Data Center at sites given in Table 1. In this study, the observation periods were classed as warm (summer

and autumn) and cold (winter) seasons to investigate the SGPF under different meteorological conditions. Noting that October-November in Beijing and May in Nanjing were categorized as cold and warm seasons, respectively.

Table 1. Information of observations for aerosol components and particle number concentrations.

| Site name | Longitude (°) | Latitude (°) | Observation period | Observed species |
|---|---|---|---|---|
| Aerosol observation | | | | |
| Beijing | 116.37 | 39.97 | 2014. 6.3-2014.7.8 2014. 10.15-2014.11.6 | Mass concentration of SNA |
| | | | 2016.11.21-2016.12.13 | PNSD and mass concentration of SNA, BC and organic matter (OM) |
| Nanjing | 118.75 | 32.06 | 2014.5.1-2014.5.31 2014.1.1-2014.1.31 | Mass concentration of SNA |
| Wuhan | 114.28 | 30.62 | 2014. 10.1-2014.10.21 | Mass concentration of SNA |
| Meteorological observation | | | | |
| Beijing | 116.47 | 39.80 | June and October 2014 | Temperature at 2m, relative humidity at 2m and wind at 10m |
| Nanjing | 118.90 | 31.93 | January and May 2014 | |
| Wuhan | 114.05 | 30.60 | October 2014 | |

**3 Results**

**3.1 Evaluation of the simulated meteorological fields**

The SGPF is closely related to meteorological conditions, especially solar radiation and wind. Stronger radiation leads to more rapid gas-phase reactions, which means more SGPF. Wind speed and direction control plume diffusion and

particle transportation in the grids. Furthermore, the concentrations of gases and aerosols in the air are related to temperature, RH, boundary layer height and other meteorological factors. We therefore compared the simulation of several meteorological factors with ground observations in corresponding cities during the same period as shown in Fig. 3. The correlation coefficient (R) of meteorological

factors are given in Table 2. The simulation of temperature and RH agreed well with observations, with correlation coefficients (R) of >0.8. Regarding wind field, the



model displays more uncertainties, with R values of 0.30–0.73 for wind speed and <0.30 for wind direction in Beijing and Wuhan. Particularly, there is an obvious descrepancy between modeled and observed variations in Wuhan, with a relatively

lower R value of 0.34. This may lead to a poor performance in modeling of aerosol distribution.

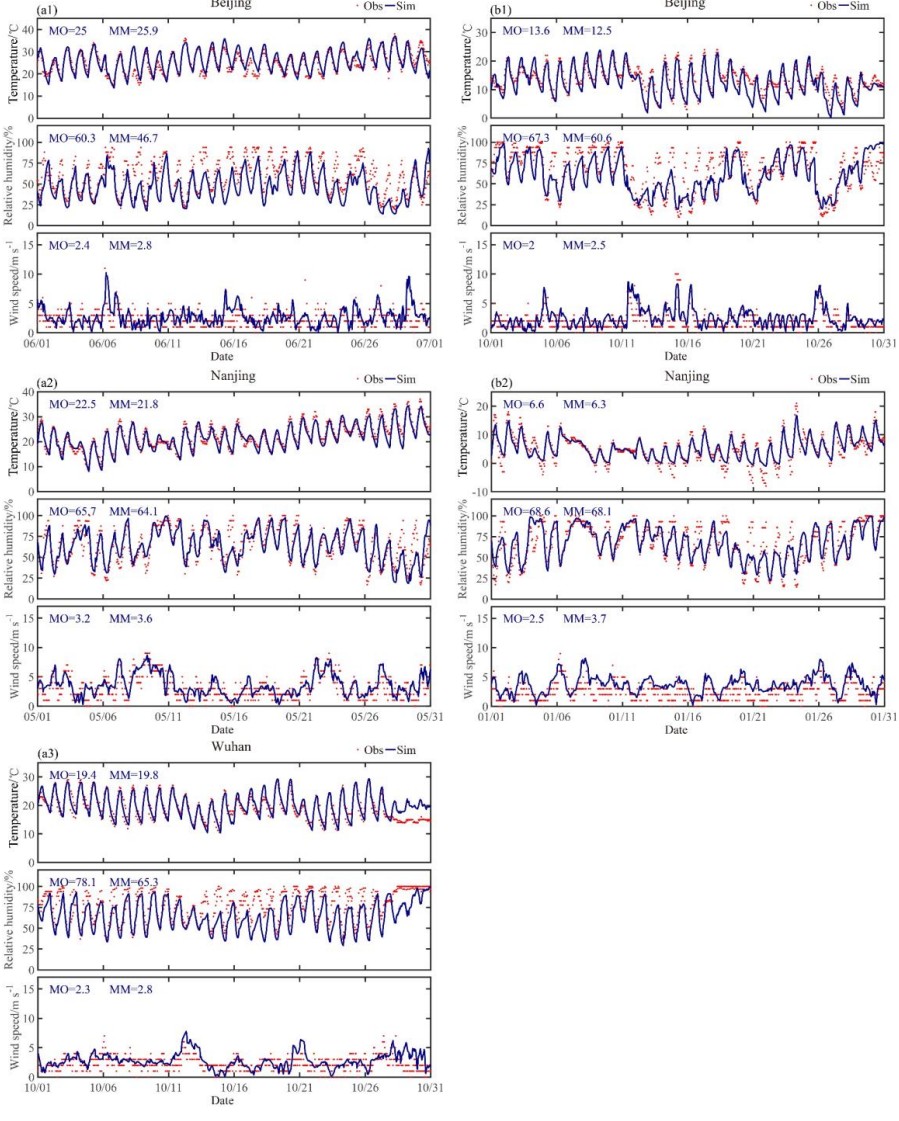

Fig. 3 Comparison between observed and simulated meteorological factors (i.e., temperature, relative humidity and wind speed) in: (a1) Beijing; (a2) Nanjing; (a3)





Wuhan in warm seasons and (b1) Beijing; (b2) Nanjing in cold seasons of 2014. The red dots and blue lines represent hourly observations and simulations, respectively. Monthly mean values of the observation and simulation are also given on subplots in blue texts, abbreviated to MO and MM, respectively.



Table 2 Summary of correlation coeficients for hourly meteorological factors in different cities. RH, U and V represent relative humidity, U wind and V wind, respectively.

| Meteorological | Warm seasons | | | Cold seasons | |
|---|---|---|---|---|---|
| factors | Beijing | Nanjing | Wuhan | Beijing | Nanjing |
| Tempreture | 0.84 | 0.94 | 0.85 | 0.88 | 0.88 |
| RH | 0.78 | 0.82 | 0.77 | 0.76 | 0.82 |
| U | 0.06 | 0.65 | -0.18 | 0.60 | 0.48 |
| V | 0.19 | 0.48 | 0.22 | 0.52 | 0.30 |
| Wind speed | 0.30 | 0.73 | 0.34 | 0.62 | 0.58 |

### 3.2 Diurnal variation of sub-grid particle formation

The diurnal variation of oxidation rate and mass concentration of SG-ASO4 is
shown in Fig. 4, with a clear diurnal pattern for $f_{ox}$ at different lattitudes (Fig. 4a, c);. with initial upward then downward trends from 08:00 to 20:00 (Beijing Time) in summer, with $f_{ox}$ varies respectively from ~0% to over 10% and from 0% to 14% in Beijing and Wuhan, respectively, due to the variation of solar radiation. At night, $f_{ox}$ remained at almost 0% due to the much lower OH concentration. In winter, $f_{ox}$
increased to ~5% during daytime and fell to ~0% at night in Wuhan, with the time for $f_{ox}$ being >1% narrowing to 10:00–18:00. For the same period, the $f_{ox}$ in Wuhan is about 1.5 times that of Beijing, and within a city, the maximum $f_{ox}$ in summer is ~3 times that in winter. Particularly, The maximum $f_{ox}$ in summer is about 6 times and 4 times that of the simulation of F2.5 for Wuhan and Beijing, respectively. Overall,
there is an evident discrepancy for $f_{ox}$ in the spatial-temporal distribution. The rate of sulfate formation at noon is significantly higher than at other time, leading to more sulfur deposition (Xu et al., 2014).

The diurnal variation in the mass concentration of SG-ASO$_4$ associated with different $f_{ox}$ and emission flux is shown in Fig. 4b, d. The modeling results of
SG-ASO$_4$ in Beijing ranged between 0–2 and 0–1 μg m$^{-3}$ in summer and winnter, respectively. In Wuhan, the maximum value exceeded 10 μg m$^{-3}$ and ~4 μg m$^{-3}$ in summer and winnter, respectively. This indicates that SFPF is substantially important in areas with large point sources. On the other hand, simulation with fixed $f_{ox}$ (2.5%)





maintained a constant SG-ASO$_4$ concentration of about 0.5–0.9 µg m$^{-3}$ and 2 µg m$^{-3}$

in Beijing and Wuhan, respectively. It is worth noting that, in Beijing, the model

results of SG-ASO$_4$ with fixed $f_{ox}$ is slightly higher in winter than summer, due to the

higher emission flux of SO$_2$ in heating season. However, the simulation with the

SGPF scheme suggests the opposite due to higher SG-ASO$_4$ conversion in summer,

demonstrating the superiority of the scheme in resolving temporal heterogeneity of

SGPF.

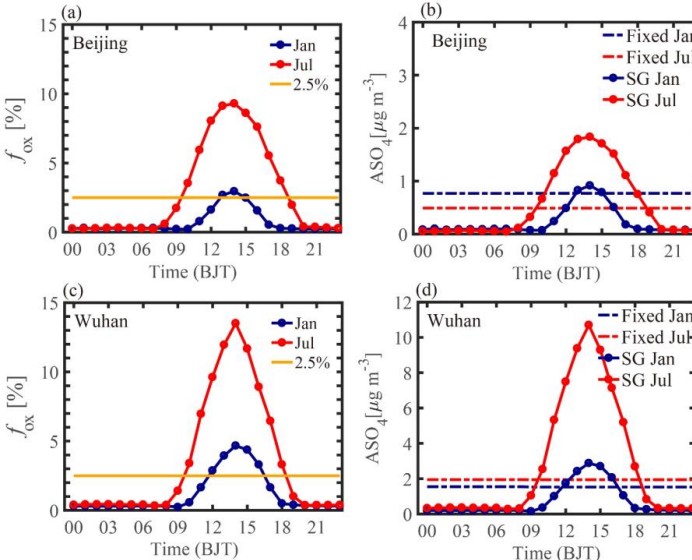

Fig. 4 Monthly averaged diurnal variation of the oxidation rate ($f_{ox}$, a, c) and mass

concentration (µg m$^{-3}$) of sub-grid sulfate (b, d) in January (blue line) and July (red

line) from the IAP-AACM. The orange line represents constant $f_{ox}$ of 2.5%. The solid

line and dash-dotted line represent simulation with the sub-grid scheme (SG) and

constant $f_{ox}$ of 2.5% (F2.5), respectively. The top row and bottom row are for Beijing

and Wuhan, respectively.

### 3.3 Improvement in aerosol components simulation

The testing of the simulated time-series of SNA mass concentration against site

observations in warm seasons is illustrated in Fig. 5, which displays the IAP-AACM

simulation with and without the SGPF scheme. The statistics for different model

results are given in Tabel 3. The model reproduced SNA concentration and their



temporal evolution well, with R values generally over 0.5. Overall, the model underpredicted secondary inorganic aerosols in different degrees. The simulation in

Beijing matched observations most closely, likely due to the more accurate emission inventory available for that area.

The SG simulation indicates sulfate increases of 1–5 μg m$^{-3}$ in warm seasons (Fig. 5). Compared with the F0 simulation, the SG simulation leads to an obvious improving on model bias (shown in Tabel 3), on the background of underestimation

of secondary inorganic aerosols in the model. The normalized model bias (NMB) in Nanjing and Wuhan were narrowed from –41% to –18% and from –60% to –33% respectively. The overestimation of the Beijing NMB increased by 20% (it was overestimated by 6% in F0). The simulation of SG significantly improved the correlation of sulfate in Nanjing with the R increasing by 0.13. The SG correlation is

similar to that of F0 in Beijing, but lower in Wuhan where the R value decreased by 0.15, with this being related to some extent to the poor wind simulation in these areas (as shown in Table 2). The SGPF scheme also has an obvious impact on the simulation of ammonium, with the NMB narrowing by 14% and R increasing by 0.08 in Nanjing and the NMB decreasing by 12% in Wuhan. The correlation of

nitrate in SG is similiar to that in the simulation of F0, with the model bias increasing by 1–9%. Overall, the IAP-AACM exhibits good performance with SNA, and the SGPF scheme improves the simulation on sulfate and ammonium.

The SGPF scheme significantly narrowed the gap between model and observation in Wuhan, with the NMB decreasing from –0.6 to –0.33 due to the high emission

rate of the point source. We found that the sulfate concentration simulated by the IAP-AACM with the SGPF scheme increased significantly during daytime, especially on October 1 and 15. The concentration of sulfate was up to 20 μg m$^{-3}$ higher on those days than the model results without SG-ASO$_4$. This could be related to the influence of the local wind direction on the plume spreading and accumulation

of pollutants. We compared the average diurnal variation of simulated SNA concentrations in Wuhan with observations in Fig. 6. The simulated diurnal profiles of both SG and F0 reproduced the variation of sulfate proportions in SNA



(ASO4/SNA) as a unimodal. Simulation with the SGPF scheme provides diurnal SNA mass concentration profiles more similar to observations than that of F0. The averaged SNA concentration was increased by ~5 μg m$^{-3}$ at night and 10 μg m$^{-3}$ in the afternoon. The simulated ASO4/SNA ratio of SG increased by 5%–10% or more at daytime and about 5% at night than F0. The simulated ASO4/SNA ratio droped much more dramatically than observed one at noon due to over-decomposition of nitric acid under high-temperature conditions.

The simulaiton of SNA in cold seasons is shown in Fig. 7. In general, the mass concentration of sulfate was obviously underestimated, with the SG simulation showing weaker improvement in the negative bias than in warm seasons. As shown in Tabel 3, the NMB of sulfate decreased by 0.16 and 0.05 in Beijing and Nanjing, respectively. The correlation increased slightly (by 0.02–0.04) due to the favorable performance of wind simulation (Table 2). The SGPF scheme slightly improved the simulation of ammonium but had ambiguous effect on the simulation of nitrate. The NMB of ammonium was reduced by 0.06 in Beijing but not changed in Nanjing as the SGPF is weakest in January. For nitrate, the simulation of SG gave little changing in correlation but increased the NMB by 0.05 in Nanjing. The evaluation of SNA simulations in SG and F0 indicates that the SGPF scheme improves model performance for sulfate and ammonium, especially in warm seasons. In particular, the consideration of SG-ASO$_4$ has a great influence on the spatial-temporal distribution of SNA near sulfur-rich stacks on condition that the simulation of wind field is reliable.

As displayed in Fig. 8, sulfate accounted for over 40% of SNA concentration in warm seasons and about 20%–30% in cold seasons in 2014. In Nanjing and Wuhan in warm seasons, sulfate from the modeling results without SGPF made up only ~30% of SNA. The simulation of SG raised the simulated proportion of sulfate by ~10% in warm seasons and <5% in cold seasons. We conclude that the model coupled with the SGPF scheme deliveres superior performance in determining the concentration of sulfate and its proportion in SNA.



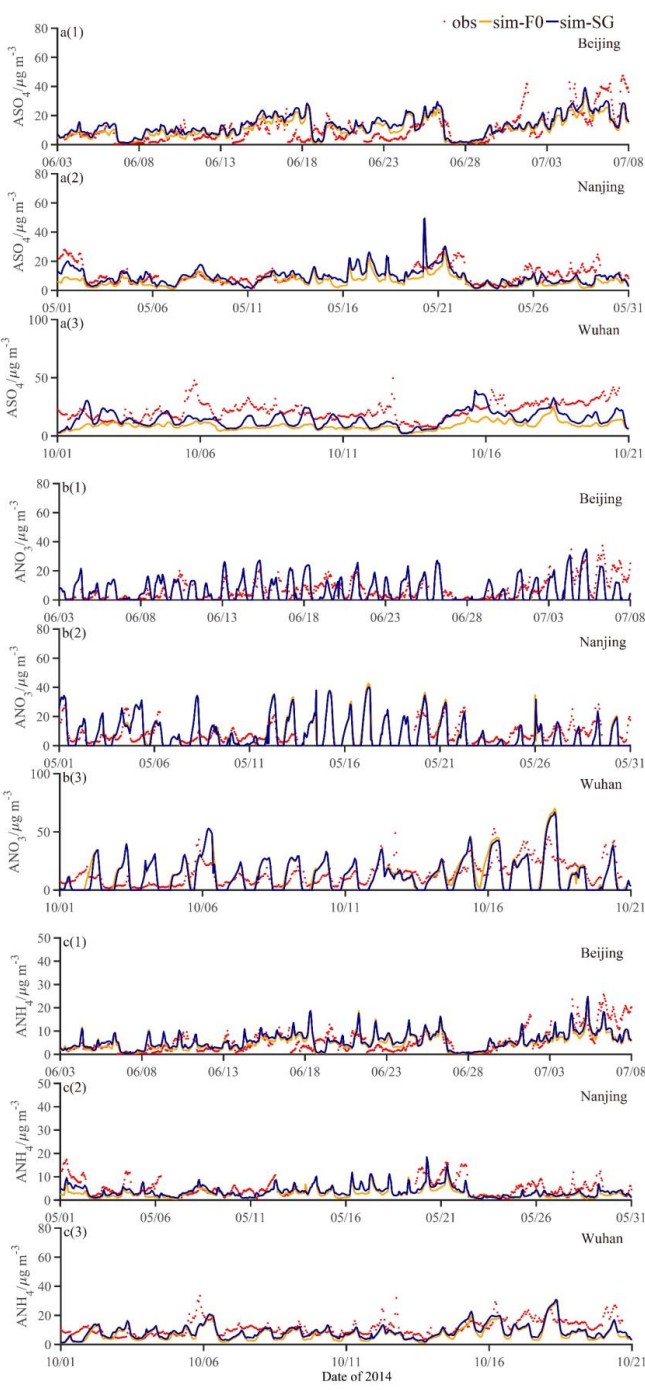

Fig. 5 Comparison of hourly simulated aerosol components against site observations

in Bejing, Nanjing and Wuhan in warm seasons: (a1–a3) sulfate, (b1–b3) nitrate, (c1–



c3) ammonium. The blue and orange line represent simulations with SG scheme and with constant $f_{ox}$ of 0% (F0), respectively. The red dots represent observations.

Table 3 Summary of statistics for hourly mass concentration of aerosol components in different seasons. MO, MM, NMB, and R represent mean value of the observations, mean value of the model, normalized mean bias, and correlation coefficients respectively. ASO4, ANO3 and ANH4 represent sulfate, nitrate and ammonium, respectively.

| Period | Site name | Species | MO ($\mu gm^{-3}$) | SG | | | F0 | | |
|---|---|---|---|---|---|---|---|---|---|
| | | | | MM ($\mu gm^{-3}$) | NMB | R | MM ($\mu gm^{-3}$) | NMB | R |
| Warm seasons | Beijing | ASO4 | 10.30 | 12.99 | 0.26 | 0.59 | 10.95 | 0.06 | 0.60 |
| | | ANO3 | 6.84 | 5.94 | -0.13 | 0.46 | 6.03 | -0.12 | 0.47 |
| | | ANH4 | 5.41 | 5.72 | 0.06 | 0.56 | 5.01 | -0.07 | 0.55 |
| | Nanjing | ASO4 | 10.72 | 8.84 | -0.18 | 0.61 | 6.33 | -0.41 | 0.48 |
| | | ANO3 | 8.58 | 7.00 | -0.18 | 0.48 | 7.47 | -0.13 | 0.50 |
| | | ANH4 | 5.36 | 3.43 | -0.36 | 0.60 | 2.69 | -0.50 | 0.52 |
| | Wuhan | ASO4 | 21.53 | 14.34 | -0.33 | 0.31 | 8.68 | -0.60 | 0.46 |
| | | ANO3 | 14.52 | 13.05 | -0.10 | 0.53 | 14.37 | -0.01 | 0.54 |
| | | ANH4 | 11.60 | 8.84 | -0.24 | 0.48 | 7.47 | -0.36 | 0.50 |
| Cold seasons | Beijing | ASO4 | 10.53 | 9.43 | -0.10 | 0.57 | 7.79 | -0.26 | 0.53 |
| | | ANO3 | 20.60 | 18.18 | -0.12 | 0.52 | 17.89 | -0.13 | 0.54 |
| | | ANH4 | 8.77 | 7.49 | -0.15 | 0.58 | 6.91 | -0.21 | 0.58 |
| | Nanjing | ASO4 | 23.65 | 9.09 | -0.62 | 0.65 | 7.91 | -0.67 | 0.63 |
| | | ANO3 | 29.41 | 18.02 | -0.39 | 0.62 | 19.28 | -0.34 | 0.62 |
| | | ANH4 | 19.82 | 7.03 | -0.65 | 0.66 | 6.91 | -0.65 | 0.63 |

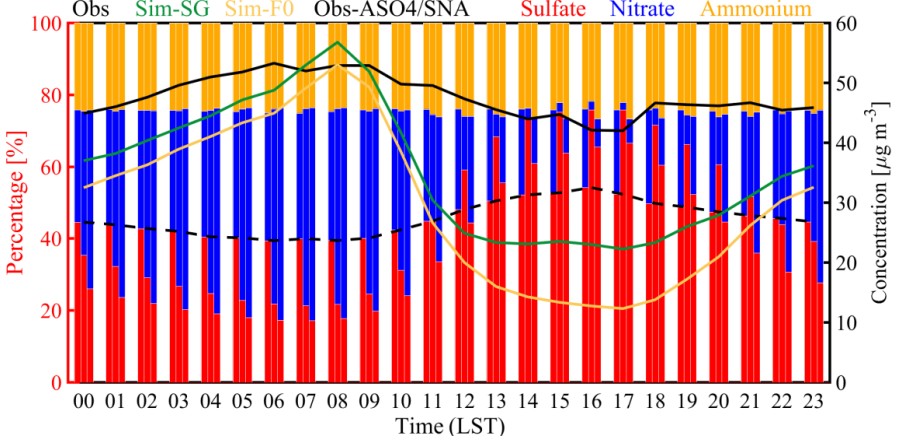



Fig. 6 Averaged diurnal variation for the observed and simulated total SNA concentrations (solid lines), observed ASO4/SNA ratio (black dot line) and the
percentage of sulfate, nitrate and ammonium in total SNA concentrations (color bars, i.e., observations, simulations of SG and F0) in Wuhan, October 1-21, 2014.

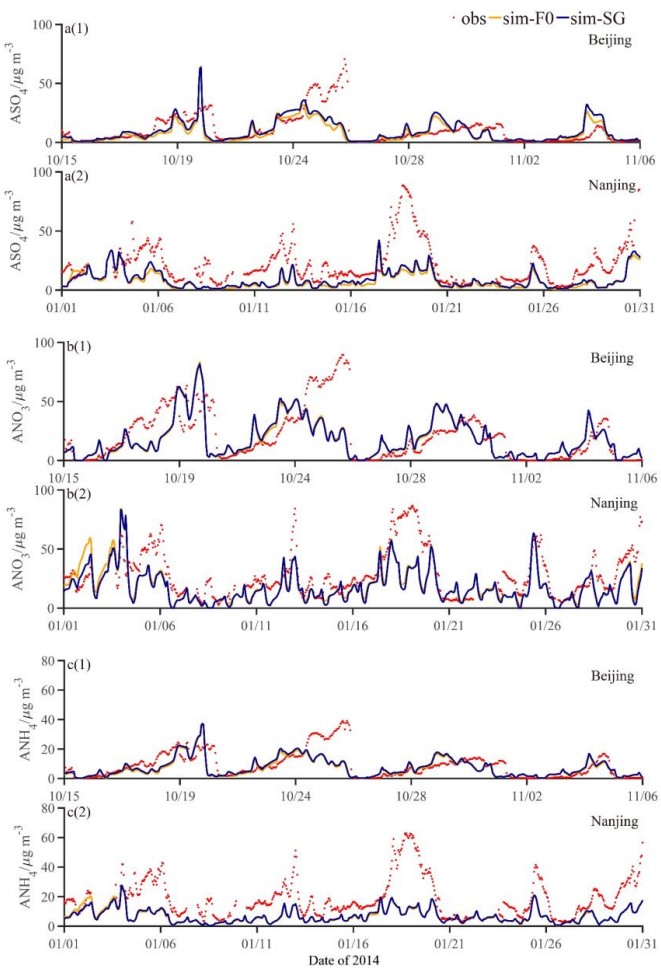

Fig. 7 The same as Fig. 5, but for Beijng and Nanjing in cold seasons.

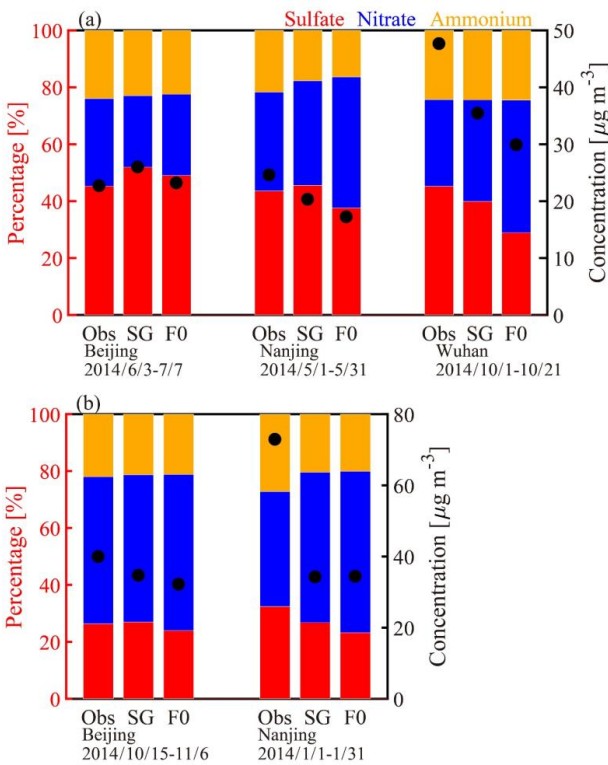

Fig. 8 Mean percentage of the simulated (SG and F0) and observed sulfate, nitrate
and ammonium (i.e., red, blue and orange color bars) and averaged SNA
concentration (black dots) in Beijing, Nanjing and Wuhan in (a) warm seasons and
(b) cold seasons.

### 3.4 Improvement in PNSD simulation

Besides the mass concentration of component, particle number concentration is an
essential parameter of aerosol particles in evaluating their climatic and environmental
effects. Most aerosol models use constant fractions for $f_{ox}$ and $f_{new}$ to describe the
SGPF, causing large uncertainties in simulating particle formation processes and particle
number concentration. To investigate the influence of SGPF scheme on particle

number concentration, we undertook two experiments using IAP-AACM+APM with
a fixed scheme ($f_{ox}$=2.5% and $f_{new}$ = 5%) and the SGPF scheme. Nodeling results
were tested against observations at the site in Beijing during the APHH-Beijing,
winter 2016. The evaluation of SNA against observations is shown in Fig. S3. The





model reproduced the mass concentration of SNA well, with R ranging from 0.59 to
0.66 and NMB ranging from –0.68 to 0.01. The PNSD of the two experiments is
shown in Fig. 9a and Fig. 9b. The observed PNSD during the same period is shown
in Fig. 9c. The model with constant $SG\text{-}ASO_4$ formation over-predicted the number
of particles in Aitken mode at night. In fact, nucleation is negligible at night due to
very low OH concentration in the plume. Compared with observations, the sub-grid
scheme significantly optimized the overestimation of particles number concentration
in small size bins. Specifically, the diurnal variation of nucleation process reduced
the positive bias by more than 3 times in nucleation mode and by about a half in
Aitken mode (see in Table 4). This indicates that IAP-AACM coupled with the
SGPF scheme reasonably captured the process of new particle formation in clean
periods (e.g. November 27–28, December 5–6) and growth in haze periods (e.g.
November 28–30). Here we also display the simulations of full spectrum of PNSD in
F2.5 and SG in Fig. S4. Particles in the simulation with F2.5 (Fig. S4a) were
produced continuously within the particle size range of 3-10nm, due to constant
nucleation of SG-ASO4. To avoid this unrealistic particle formaiton, the fraction of
$SG\text{-}ASO_4$ is always set as 0% in aerosol models. However, it will result in a
considerably underestimation in nucleation mode and Aitken mode. Taking into
account the influence of solar radiation on SGPF, the model with the SGPF scheme
reproduces the diurnal cycle of particle number in small-sized bins and provides
more reasonable numbers concentration of CCN.



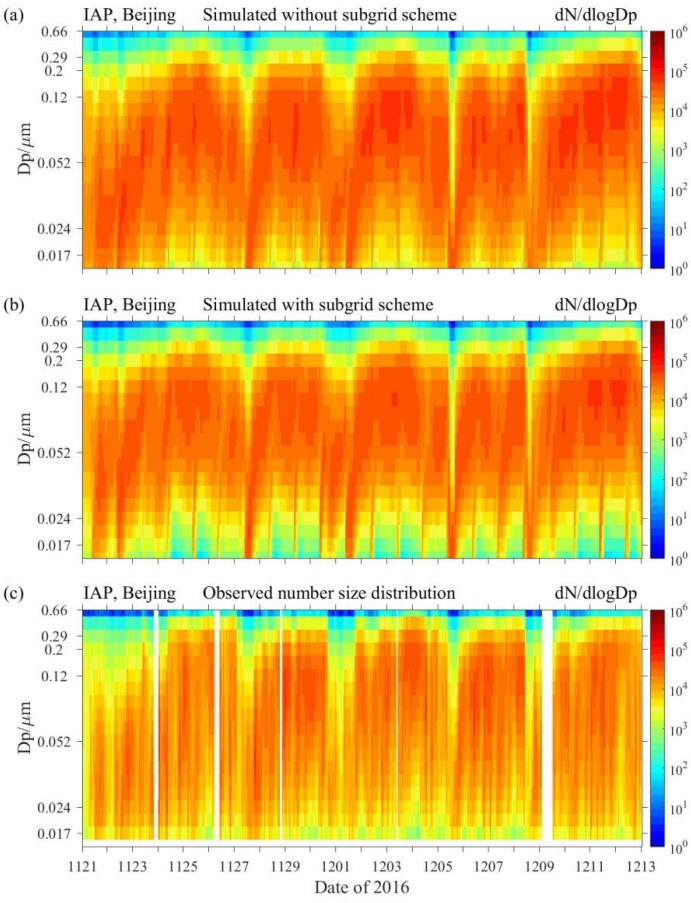

Fig. 9 Particle number size distribution of simulations from IAP-AACM with (a) F2.5 and $f_{new}$ = 5%, (b) SG and (c) observations during the APHH-Beijing in 2016.

Table 4 Mean number concentrations of the observations and simulations and the NMB for different modes at the site in Beijing during the APHH-Beijing in 2016.

| Experiments | Nucleation | | Aitken | | Accumulation | |
|---|---|---|---|---|---|---|
| | Con(cm⁻³) | NMB(-) | Con(cm⁻³) | NMB (-) | Con(cm⁻³) | NMB (-) |
| Observation | 1312 | | 10223 | | 3754 | |
| SG | 2208 | 0.68 | 15183 | 0.49 | 7022 | 0.87 |
| F2.5 | 4412 | 2.36 | 18507 | 0.81 | 7160 | 0.91 |

**3.5 Regional impacts of sub-grid particle formation**





The significant impact of SGPF on sulfate concentration near site is shown in Fig. 5. The distribution of sub-grid particles in local areas is influenced strongly by wind field due to dilution of high plume. To explore the spatial inhomogeneity in regions around point sources, the regional impact of SGPF during typical periods is illustrated

in Fig. 10. In Nanjing and Wuhan, the area represented by the observation site is located downwind of the point source when the easterly wind prevails. For Nanjing, there is then an incease in sulfate mass concentration by 25%–50% around the observation site, in Wuhan, reaching reaching >50% in the downwind area. This should be ascribed to the high capacity of local power plants with SO$_2$ emission of

over 2 μg m$^{-2}$ s$^{-1}$ in suburban Wuhan, indicating the significant impact of point source with high emission rate on both the spatial scale and material concentration of the aerosol distribution. On the other hand, the regional influence in Bejing was relatively small (Fig. 10a, d), due to the upwind location and much lower emission rate of plants. Comparison of the three scenarios indicates that, although sub-grid particles from

elevated sources are discharged to high altitudes, they can still contribute significantly to local ground pollution under unfavorable wind condition.

The influence of the SGPF diurnal profile (Section 3.2) on regional scales was investigated at two particular times (i.e., 14:00 and 02:00 Beijing time). To exclude the influence of boundary layer height on aerosol concentration, we used sulfate/BC

ratios to nornalize the comparison between day and night. As shown in Fig. 11, in industrial areas in eastern China there is a larger increase in sulfate concentration due to SGPF in daytime than at night. The increase in large areas of Shanxi Province indicates the important role of coal burning as the main energy source. In central China, there are several maxima which indicates the local influence of SGPF. Note

that the highest increase occurred in northwest China because the concentration of BC is almost zero in such clean areas. Overall, the temporal variation of SGPF affect the sulfate concentration on a large-scale in eastern China, especially in areas with high-emission sources, leading to heterogeneous spatial distribution.

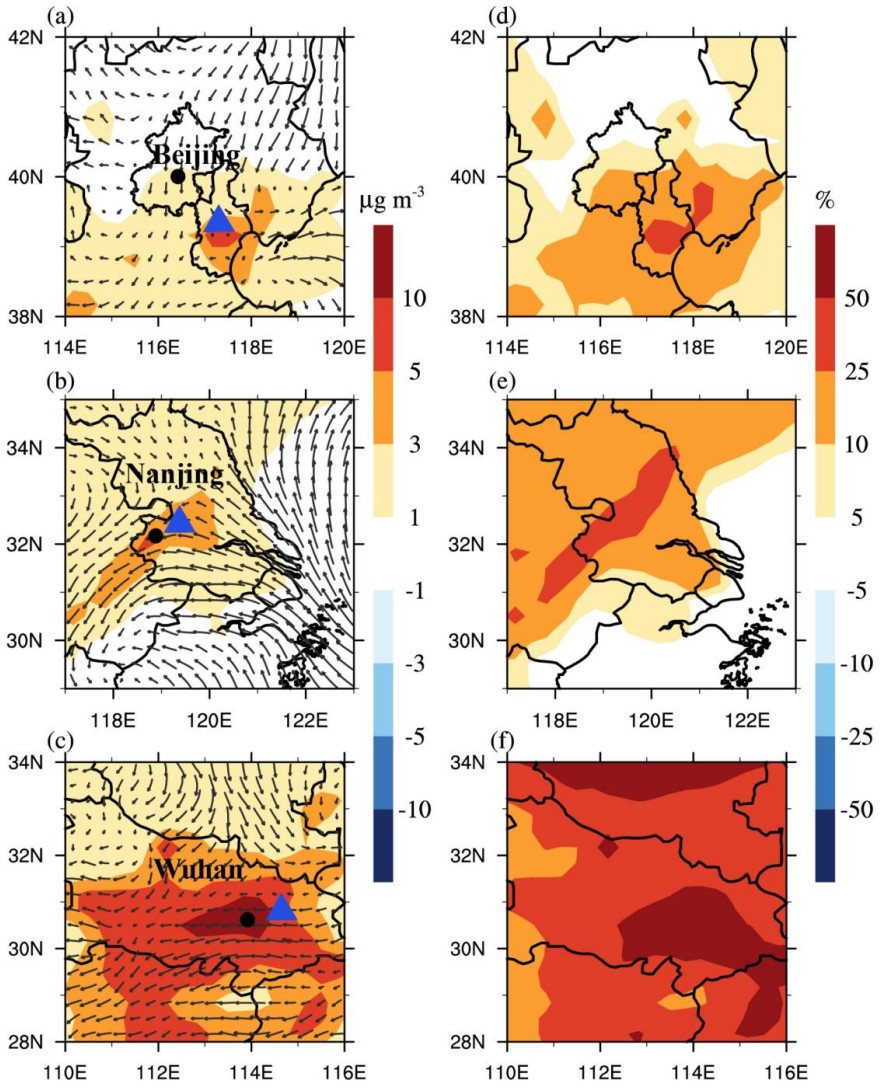

Fig. 10 Daily averaged differences with overlayed wind filed (left column, µg m⁻³) and relative change (right column, %) of sulfate concentrations on regional scale between SG and F0 at surface, around: (a, d) Beijing in June 22, (b, e) Nanjing in May 16 and (c, f) Wuhan in October 15. The difference was calculated as the simulation of SG minus F0. Black dots and blue triangles represent locations of the ground sites and

plants, respectively.

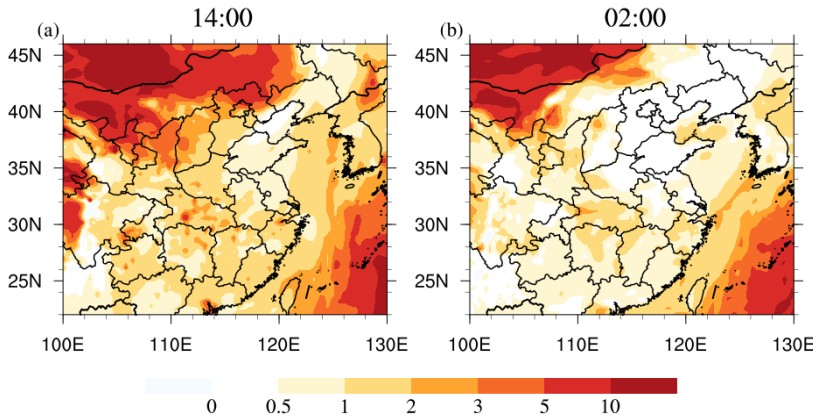

Fig. 11 Difference of sulfate/BC at July 1st (a) 14:00 and (b) 02:00 Beijng time
between SG and F0 at surface. The difference is calculated as the simulation of SG
minus F0.

**4. Conclusions and discussions**

In this study, the P6 sub-grid parameterization scheme was coupled with the
global nested aerosol model IAP-AACM to resolve the SGPF characteristics in both
oxidation and nucleation processes over eastern China. Furthermore, the key
parameter of the scheme, effective OH concentration in the plume, was modified to fit
the local chemical background on the basis of extensive field observations in eastern
China, on account of the dependence of radicals on precursor gases. With the updated
model integrated with the localized new scheme, the spatio-temporal inhomogeneity
of SGPF was well parameterized in the grids. The significance of sub-grid particles in
simulating the distribution of secondary inorganic aerosols was demonstrated in
comparisons against site observations for both aerosol mass concentration and PNSD
in different seasons.

The major findings were as follows. (1) The SGPF scheme provides a more
reasonable description of oxidation and nucleation processes for SGPF, including the
diurnal variation of $f_{ox}$ and $f_{new}$. The spatio-temporal inhomogeneity of sub-grid
particles distribution is well characterized. The spatial and temporal discrepancy is
obvious, as for the same area the $f_{ox}$ in summer can be three times that of winter. (2)
simulations with the SGPF scheme better reproduce nanoparticle formation than the



fixed oxidation fraction as the physicochemical parameterization profiles more accurately describe the nucleation source at night. The overestimation of particle number in small size was reduced by over 3 times in nucleation mode and roughly one half in Aitken mode. (3) The SGPF gave more significant improvement in model performance in warm seasons as it reduced the NMB of sulfate and ammonium by 12%–27% at most sites, while increasing that of nitrate by 1%–9%. In cold seasons, the influence of SGPF was limited, with the NMB for sulfate decreasing by only 5%–16%. In cold seasons, the influence of SGPF is limited with the NMB for sulfate only reduced by 5%–16%. The simulating improvement on correlation will experience a significant increase by 0.13 if the model performance of wind fields are nice. (4) The SGPF scheme can improve model performance in determining the concentration of sulfate and its proportion in SNA, increasing the proportion of sulfate by ~10% in warm seasons and <5% in cold seasons. Specifically, in Wuhan, the averaged SNA concentration increased by ~5 μg m$^{-3}$ at night and 10 μg m$^{-3}$ in the afternoon, which has implications for sulfate simulating in areas near stacks. (5) The SGPF has a significant impact in local areas near point sources, with the sulfate concentration increasing by 25%–50% or even >50% under downwind conditions. This indicates that the impact of SGPF should be taken into account in studies of air pollution and aerosol formation in extensive industial cities, not only in China, but also in other developing countries.

As the IAP-AACM was driven by the global WRF off-line, the aerosol feedback was not taken into account in this work. The SGPF scheme didn't contain the variation in OD associated with droplets in the plume. Without the aerosol-radiation interaction (ARI), the impact on increasing atmospheric stability by cooling the surface but heating the air aloft was excluded. The impact of aerosols serving as cloud condensation nuclei on optical properties and lifetime of clouds and precipitation was also not included. Moreover, the scattering and absorption of ultraviolet radiation by aerosols influence the photolysis rates and reduce the formation of O$_3$ and other oxidants (He and Carmichael, 1999). In high polluted area, ignoring the impact of OD will cause a certain degree of overestimation of atmospheric oxidation capacity. Li et





al. (2018) incorporated recently reported heterogeneous chemical mechanisms into the regional version of IAP-AACM (NAQPMS) and found that perturbations in

photolysis frequencies reduced $O_3$ concentrations by 1–5 ppb in winter and 1–3 ppb in summer. Considering both the ARI and the aerosol–photolysis interaction (API), Wu et al. (2020) conducted model experiments in North China Plain and found that ARI contributed to a 7.8% increase in near-surface $PM_{2.5}$, while API suppressed secondary aerosol formation to a 3% decrease of $PM_{2.5}$. Therefore, the overestimation of

atmospheric oxidation capacity caused by ignoring the OD impact should be under 5% in eastern China. To better understand the impact and uncertainty of sub-grid particle, the impact of OD should be included in the model in our future work.

**Code and data availability**

All of the observation data needed to evaluate the conclusion of this paper are

provided in the main text. The source codes of IAP-AACM with SGPF scheme are available online via ZENODO (https://doi.org/10.5281/zenodo.4383361; Wei et al., 2020). Please contact Ying Wei (weiying@mail.iap.ac.cn) to obtain the model data for IAP-AACM.

**Author contribution**

YW developed the model, did the simulation, and wrote the paper. XC developed the model and designed the study. HC developed the model and prepared the gridded emission data. YS provided the observation data. WY, HD, QW, DC, XZ, JL and ZW modified the paper. YW and XC prepared the manuscript with contributions from all the co-authors.

**Acknowledgments:**

We sincerely thank Prof. Fangqun Yu at State University of New York at Albany for providing the code of original APM box model. We also thank the Jiangsu Environmental Monitoring Center and Hubei Environmental Monitoring Center for their supports with aerosol composition data of Nanjing and Wuhan respectively. This

research is supported by the National Key R&D Program of China (Grant NO. 2020YFA0607801), the National Natural Science Foundation of China (Grant NO. 42007199; 41907200) and the National Key Scientific and Technological





Infrastructure project "Earth System Science Numerical Simulator Facility".



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
