# Peer review of "Investigating the importance of sub-grid particle formation in point source plumes over eastern China using IAP-AACM v1.0 with a sub-grid parameterization"

_Geoscientific Model Development, 2020_

## Author Comment (AC2)

**We thank the reviewers for the effort to review the manuscript and to provide constructive comments and good suggestions to improve our manuscript. Our replies to the comments and our actions taken to revise the paper (in blue) are given below (the original comments are copied here).**

The modifications corresponding to the comments and the revised language and grammars in the manuscript are marked in red color.

Referees' comments:

Referee #1

Oxidation process of SO2 to sulfate is a key factor influence atmospheric aerosol particle chemical composition, size distribution, new formation. This can largely impact the hygroscopicity and available cloud condensation nuclei in the atmosphere and therefore lead to impacts on atmospheric chemical processes and climate. However, this process is usually with a scale much less than regional or climate models' resolution, and the sub-grid oxidation process could be an important source of uncertainty and hamper our better understanding in air pollution and climate. This study took a further step, by including a sub-grid scheme to describe this sub-grid process in a global-regional model to quantify the uncertainty introduced by sub-grid oxidation of SO2 and improve the model performance. I think this is a good piece of work and well fit the scope of GMD, in terms of science. But, I also notice there are many typos and ambiguous statements in the manuscript. I would like to suggest more attention and carefulness on the language and presentation. In general, I believe this work is worth for publishing in GMD after some minor corrections and careful language editing.

Reply: Thanks to the reviewer for detailed review and good suggestions to improve our work.

Specific comments:

1) end of page-2, coal burning in China contribute 80% SO2 emission. Do you mean 80% of china total emissions or global emissions? And, at nowadays, SO2 emission in

China has effectively reduced due to the great success of green energy policy of China. But India surpass the emission of China and tops the SO2 emissions (Li et al., 2017). This point also worth to comment on.

Reply: "coal burning in China contribute 80% SO2 emission " means coal burning contribute 80% to SO2 emission in China. As pointed out by the reviewer, in China, SO2 emissions have been effectively reduced due to the desulfuration measures and reduction policies. However, the coal burning is still the major SO2 sources. On the other hand, the SO2 emissions of India has surpassed that of China. So the study of sub-grid parameterization on point source can also help to make sense of the air pollution problem in India reported in recent studies (Chen et al., 2020). Some references on SO2 emissions and air pollution in India were added and the sentences about this issue have been made more clear to avoid misunderstanding (see in Line 57-62 and Line 649-650).

Chen, Y., Wild, O., Ryan, E., Sahu, S.K., Beig, G.J.A.C., Physics, 2020. Mitigation of PM2.5 and ozone pollution in Delhi: a sensitivity study during the pre-monsoon period. 20, 499-514.

2) line 64-66, please double check the chemical equation. I believe it is OH radical, rather than anion. It would be better to explain that why the concentration NOx and VOCs will affect the oxidation of SO2.

Reply: Yes, OH is radical rather than anion. The chemical equation is revised to "2OH+SO2→H2SO4". The concentration of NOx and VOCs can influence the atmospheric oxidation though gas-phase chemical reactions and thus the OH concentration and the oxidation of SO2. According to this good comment, the explanation is added (see in Line 64-66).

3) lots of typos, here are some examples, but I believe there are many more. Please carefully check the manuscript. Line-71, primary? I think should be 'secondary'?; line 93-94: tens of seconds of kilometers, I do not understand here; 'caocentration'; line 279: 'imparct', etc…

Reply: As the process of H2SO4 nucleating to form new particles in the plume is much faster than the formation through the gas-phase reaction in the atmosphere, the new formed sulfate is named as 'primary' (Luo and Yu, 2011). 'caoncentration' should be 'concentration'. 'imparct'' should be 'impact'. Other typos have also been

carefully revised (seen in line 98, 286, 405, 455, 536, 585, etc.).

*Luo, G. and Yu, F.: Sensitivity of global cloud condensation nuclei concentrations to primary sulfate emission parameterizations, Atmospheric Chemistry and Physics, 11, 1949-1959, 2011.*

4) please provide the units for all variables in your Eq. 1-6. And you have two equation 5 and 6. What is DSWRF in your Eq. 6? Downward shortwave at TOA or surface?

Reply: The units for all variables in Eq. 1-6 are provided except dimensionless variables (e.g., P1 and P2) (see in Line 237, 239 and 265). The numbers of equations are revised in Line 264 and 268. DSWRF is the downward shortwave radiative flux from the WRF model. The explanation of DSWRF is shown in Line 209.

5) in the P6 scheme, 'x' is calculated as a function of [NOx] depend on high/low-VOC regime. In a very intensive plume, high concentration of fresh emitted NOx would deplete oxidants. Would you please make some comments on this, and discuss how could this effect influence the results.

Reply: The depletion of oxidants in the high NOx concentration plume is considered in the SGPF scheme. As shown in Fig. 1, the OH concentration decreases as the NOx concentration increases when the NOx concentration is higher than ~5ppb in the localized curve. The OH concentration falls to ~$2 \times 10^6$ cm$^{-3}$ when NOx concentration is higher than 30 ppb. Before localization, the OH is depleted more when the NOx is high. The NOx–OH curve without localization fits to the atmospheric condition in Europe and America. However, the NOx and OH concentration are much higher in the atmosphere of China. Therefore, we adjusted the variation curve of OH concentration with respect to the NOx concentration in the plume based on the surface observations. Obviously, there are uncertainties in this parameterization. The variation curve can be further updated when observations in the plume are available. Some comments have been added in Line 315-325.

6) line 300, I do not quite understand there. Why OH is in the range of (1-8)*1e6, but with a peak of 2.7*1e6? Should the peak of 8*1e6? I could be lost in somewhere, please help make it clear.

Reply: The observed daily maximum OH concentrations is in the range of $(1–8) \times 10^6$

cm$^{-3}$ and the averaged daytime peak is $2.7 \times 10^{6}$ cm$^{-3}$ over the whole observation period. It has been revised in Line 308.

7) section 2.4. The outer domain of your model is a global domain. But, as I understood, WRF is a regional model. How could a regional model drive a global domain?

Reply: Yes, WRF is more widely used in regional simulations. However, WRF also support global simulation applications (Zhang et al., 2012). The reference of global WRF is added in the revised manuscript in Line 328.

*Zhang, Y., Hemperly, J., Meskhidze, N., and Skamarock, W. C.: The Global Weather Research and Forecasting (GWRF) Model: Model Evaluation, Sensitivity Study, and Future Year Simulation, Atmospheric and Climate Sciences, 02, 231-253, 10.4236/acs.2012.23024, 2012.*

8) Would you please comments on that why the performance in Beijing warm season is worse in P6 scheme?

Reply: It should be related to the poor simulation of wind field in Beijing in warm seasons. As shown in Table 3, the correlation coefficients of both U wind and V wind of Beijing is much lower in warm season (0.06 and 0.19) than in cold season (0.60 and 0.52). Moreover, the correlation coefficients of wind speed in warm season is more than 50% lower than in cold season. The simulation of wind will directly affect the diffusion and transportation of air pollutants. Some comments are added in Line 393-395

9) line 648-650. ARI contributed to a 7.8% increase in near-surface PM2.5, while API suppressed secondary aerosol formation to a 3% decrease of PM2.5. I do not understand here. First, what is API? Second, why suppress the secondary formation but contribute to a 7.8% increase in PM2.5.

Reply: API and ARI mean the aerosol–photolysis interaction and aerosol-radiation interaction, respectively. As ARI increases atmospheric stability, it contributed to a 7.8% increase in near-surface PM$_{2.5}$ according to the experiments conducted by Wu et al. (2020). However, API suppressed secondary aerosol formation to a 3% decrease of PM$_{2.5}$. The description of the impacts of ARI and API on PM$_{2.5}$ concentrations here is to estimate the impact of not containing aerosol feedback on our results.

10) figure quality is not good, especially figure 7 and 5.

Reply: Figure 5 and 7 have been rearranged to make them more clear. However, the sharpness of the figures pasted into the document is not satisfactory due to the large number of subgraphs. We will submit *.eps images separately later.

Reference:

Li, C., McLinden, C., Fioletov, V. et al. India Is Overtaking China as the World's Largest Emitter of Anthropogenic Sulfur Dioxide. Sci Rep **7,** 14304 (2017). https://doi.org/10.1038/s41598-017-14639-8

Referee #2

This study examined the impacts of sub-grid particle formation (SGPF) in point source plumes on 20 aerosol particles over eastern China in IAP-AACM. By implementing a SGPF scheme into the model and optimizing the key parameter in the scheme, the authors found that the model performance in simulating aerosol components and new particle formation processes was improved, indicating that SGPF processes are important in chemical transport model. This study can contribute to the CTM community and the results are solid. It can be considered to be accepted after addressing my comments below.

There are two steps for improving the model in this study. First, coupling the P6 sub-grid parameterization scheme with the global nested aerosol model IAP-AACM. Second, modifying the key parameter of the scheme, effective OH concentration in the plume, to fit the local chemical background on the basis of extensive field observations in eastern China. Four simulations are performed including SG and F0 for 2014 and SG and noSG(fox 2.5?) for winter 2016. I don't get what questions were the authors trying to answer. Why did they design these two sets of simulations? Why don't they directly use SG and original model setup in all places, which should represent the improvement of the model.

Reply: We greatly appreciate the reviewer for insight comments on the manuscript. There are two groups of comparisons in the manuscript, one is between SG and F0, and the other one is between SG and F2.5. The SG experiment represented the simulation with the localized SGPF scheme. The F0 and F2.5 are without SGPF scheme, but employed fox=0% (without sub-grid particles) and fox=2.5%, respectively. The comparison between SG and F0 were conducted to evaluate the sub-grid particles' impact on aerosol mass concentration simulation. The comparison between SG and F25 were conducted to explore the impact of SGPF scheme on the model performance in PNSD. The description is added in Line 336-344 and we have added a table (see in Table 1) to describe the experiments conducted in the paper.

Specific comments:

Lines 29, 31, 35: reduced and increased from xx to xx.

Reply: As the reductions and increases in different areas were different, we used a range to represent the variations between the simulation with sub-grid scheme and without sub-grid scheme.

Line 32: Since here is the diurnal cycle, the overestimation is for a specific time or for the whole day.

Reply: The overestimation of particle number concentration is at night. The time has been added in Line 33.

Line 46: Suggest to include some recent studies (e.g., Yang et al., 2019, 2020)

Reply: Thanks a lot for your good suggestion, some recent references have been added in Line 47.

Lines 80-83: Is 0-5% of SO2 emitted as H2SO4? Is the 0-15% of H2SO4 from 0-5% of total SO2 or the 0-15% of new partial from the total H2SO4?

Reply: Yes, 0-5% of total SO2 emitted as H2SO4, and 0-15% of H2SO4 is taken as the newly formed particles through nucleation. The sentences about this issue have been made more clear to avoid misunderstanding (see in Line 84-87)

Line 93: What does the "tens seconds of kilometers" mean?

Reply: It means a spatial scale of 10s km$^{-1}$ that the gas-to-particle process is very fast, the unit has been changed to numeric description in Line 97.

Line 315: Suggest to add a table describing the detail of the simulation and what they are used for.

Reply: Thanks for the good suggestion. A table has been added to describe the experiments conducted in the paper (see in Table 1).

Line 341: Do you mean emergy and industry sectors were emitted into the first "five and three" layers of the model, "respectively"?

Reply: Yes, it is. The sentence was revised in Line 353.

Lines 343 and 345: Why the emissions in 2014 are from HTAP2 together with a scaling factor and the emissions in 2016 are directly from MEIC? MEIC also provides 2014 emissions.

Reply: The scaling factors of the emissions in 2014 are from the study by Zheng et al. (2018), and the variations of emissions during 2010-2017 in his study were based on the MEIC inventory.

Line 526: "Nodeling" to "Modeling"

Reply: Thanks, the typo has been corrected in Line 537.

Line 575: "nornalize" to "normalize"

Reply: Thanks, it has been corrected in Line 586.

Line 635:   What does the "OD" represent?

Reply: "OD" means optical depth. The explanation was shown in Line 157.

References:

Yang, Y., S. J. Smith, H. Wang, C. M. Mills, and P. J. Rasch, Variability, timescales, and nonlinearity in climate responses to black carbon emissions, Atmos. Chem. Phys., 19, 2405–2420, doi:10.5194/acp-19-2405-2019, 2019.

Yang, Y., Ren, L., Li, H., Wang, H., Wang, P., Chen, L., Yue, X., and Hong, L., Fast climate responses to aerosol emission reductions during the COVID-19 pandemic, Geophys. Res. Lett., 47, e2020GL089788, doi:10.1029/2020GL089788, 2020.

---

## Author Response (AR2)

We sincerely thank the editor and reviewers for their efforts to review the manuscript and good suggestions to improve the quality of this paper. We have carefully checked the manuscript, and the revised language and grammars in the manuscript are marked **in red** color in revision mode (see in pdf file).